# Impact of Bedside Balloon Atrial Septostomy in Neonates with Transposition of the Great Arteries in a Neonatal Intensive Care Unit in Romania

**DOI:** 10.3390/life13040997

**Published:** 2023-04-12

**Authors:** Catalin Cirstoveanu, Carmina Georgescu, Mihaela Bizubac, Carmen Heriseanu, Corina Maria Vasile, Irina Margarint, Cristina Filip

**Affiliations:** 1Department of Neonatal Intensive Care, “Carol Davila” University of Medicine and Pharmacy, 020021 Bucharest, Romania; catalin.cirstoveanu@umfcd.ro; 2Neonatal Intensive Care Unit, M.S. Curie Children’s Clinical Hospital, 041451 Bucharest, Romania; mariana-carmen.iliescu@drd.umfcd.ro; 3Ph.D. School Department, “Carol Davila” University of Medicine and Pharmacy, 020021 Bucharest, Romania; 4Department of Pediatric and Adult Congenital Cardiology, University Hospital of Bordeaux, 33600 Bordeaux, France; corina.vasile93@gmail.com; 5Department of Pediatric Cardiology, M.S. Curie Children’s Clinical Hospital, 041451 Bucharest, Romania; 6Department of Pediatric Cardiovascular Surgery, M.S. Curie Children’s Clinical Hospital, 041451 Bucharest, Romania; 7Department of Pediatrics, “Carol Davila” University of Medicine and Pharmacy, 020021 Bucharest, Romania

**Keywords:** transposition of the great arteries, Rashkind procedure, atrial balloon septostomy, echocardiography, newborn, arterial switch

## Abstract

(1) Background: Transposition of the great arteries (TGA) is the most common congenital heart disease, accounting for 5–7% of all cardiac anomalies, with a prevalence of 0.2–0.3 per 1000 live births. (2) Aim: Our main objectives were to evaluate the clinical safety of balloon atrial septostomy in neonates and the possible complications. Furthermore, we tried to establish whether the procedure should be performed in all TGA patients with small atrial septal defects, regardless of oxygen saturation, within a center where corrective surgery cannot be performed on an emergency basis due to the lack of a permanent cardiac surgery team for arterial switch surgery. (2) Methods: We conducted an observational, retrospective, single tertiary-care center study between January 2008 and April 2022, which included 92 neonates with TGA transferred to our institution for specialized treatment. (3) Results: The median age at the time of the Rashkind procedure was four days. The rate of immediate complications after balloon atrial septostomy (BAS) was high (34.3%), but most were transient (metabolic acidosis and arterial hypotension—21.8%). Twenty patients with TGA managed in our hospital underwent definitive and corrective surgical intervention (arterial switch operation) at a median age of 13 days. Most patients (82.6%) were term neonates, but 16 were born preterm. (4) Conclusions: Urgent balloon atrial septostomy is often the only solution to restore adequate systemic perfusion. Bedside balloon atrial septostomy is a safe, effective, and initial palliative intervention in neonates with TGA, which can be performed in the neonatal unit.

## 1. Introduction

### 1.1. Transposition of the Great Arteries, the Main Indication for Balloon Atrial Septostomy

Transposition of the great arteries (TGA) is the most common congenital heart disease, accounting for 5–7% of all cardiac anomalies, with a prevalence of 0.2–0.3 per 1000 live births [1].

Transposition of the great arteries (TGA) is a conotruncal anomaly in which the aorta arises from the right ventricle (RV) and the pulmonary artery originates from the left ventricle (LV), resulting in discordant ventriculo-arterial connections. This abnormality can occur isolated or be associated with other heart defects [2]. Therefore, non-oxygenated blood is recirculated into the body via the right ventricle–aorta connection. Meanwhile, oxygenated blood is recirculated to the lungs via the left ventricle–lung artery connection. 

At least two possible communications between the systemic and pulmonary circulation are required for the patient to survive: patent ductus arteriosus (PDA) and atrial septal defect—both present during fetal life, or ventricular septal defect. Therefore, transposition of the great arteries can be classified according to the presence or absence of ventricular septal defect (VSD): simple TGA with an intact ventricular septum (IVS) and complex TGA in association with other cardiac anomalies, including VSD.

Once the diagnosis of TGA is confirmed by transthoracic echocardiography, the atrial septal defect should be assessed to verify whether there is an adequate left-to-right shunt supplying oxygenated blood to the systemic circulation. In the case of a minor atrial septal defect, postnatal physiological increases in pulmonary blood flow, and the left atrial pressure may lead to the closure of the foramen ovale flap valve [3].

The hemodynamics of the interatrial shunting and PDA in Dextro-Transposition of the Great Arteries (D-TGA) with an intact ventricular septum is complex. It depends upon the relative difference between pulmonary and systemic vascular resistance and ventricular compliance. D-TGA with an intact IVS might be associated with persistent pulmonary hypertension and low pulmonary vascular resistance, leading to changes in the hemodynamics and direction of flow through the PDA and the patent foramen ovale (PFO) [4]. Atrial shunting depends on the size of the ASD and the right and left atrial pressures. The compliance of the atrial wall, the venous return, and the ventricular compliance determine atrial pressures.

If the diagnosis has been made prenatally, it is recommended to consider birth planning in tertiary-care centers with special expertise in managing complex congenital heart disease, where prompt and appropriate stabilization can be provided, followed by surgical repair and postoperative multidisciplinary management. Patients with reduced circulatory mixing (i.e., simple TGA with restrictive foramen ovale and closed ductus arteriosus) become symptomatic shortly after birth with extreme cyanosis, hypoxia, and acidosis, as well as increased cyanosis, hypoxia, acid lactate, low urine output, and eventual circulatory collapse. Left untreated, TGA leads to progressive hypoxia, acidosis, and eventual death.

The initial management of newborns with TGA focuses on stabilization, aiming to achieve adequate tissue oxygenation and arterial oxygen saturation of 75–85% (above 70% in preterm newborns) [5]. Optimization of blood mixing between the systemic and pulmonary circulations consists of continuous infusion with prostaglandin E1 (PGE1) to maintain the patency of the ductus arteriosus (PDA); balloon atrial septostomy, mild hyperventilation, and increased fraction of inspired oxygen (FiO_2_) to lower pulmonary vascular resistance and increase pulmonary blood flow; blood transfusion to increase oxygen carrying capacity; sedation and paralysis to decrease oxygen consumption; and inotropic support for increased cardiac output and oxygen delivery [6].

In some cases, using PGE1 for PDA shunting may be insufficient in restrictive interatrial communication. In these patients, atrial septostomy must be performed to allow for efficient mixing between the two circulations. Atrial septostomy is an endovascular procedure that maintains the communication between the right and left atria, increasing oxygen saturation in the systemic circulation and decreasing the left atrial pressure. The procedure is also beneficial in patients with other complex congenital heart defects, such as hypoplastic left heart syndrome, total anomalous pulmonary venous connection with restrictive ASD, tricuspid atresia with restrictive ASD, pulmonary atresia with IVS, etc. 

Atrial septostomy is especially performed in infants under six weeks of age and has certain limitations in older patients due to the increased thickness of the atrial septum. In such cases or in some congenital heart diseases (CHDs), where the interatrial septum is thicker (such as mitral valve atresia), dilation of the interatrial septum by blade septostomy or static septostomy is preferred. In this paper, we will focus on balloon atrial septostomy.

The arterial switch operation is the gold standard when the anatomical conditions and the timeline are appropriate. The optimal period for performing arterial switch operation is between the first days and up to 3 weeks of life. In Europe, surgical intervention is routinely performed during the first week of life in 25–30% of newborns [7].

### 1.2. Balloon Atrial Septostomy Technique

Atrial septostomy was first described as a surgical technique by Blalock and Hanlon in 1950. Sixteen years later, Rashkind and Miller introduced the non-surgical approach using a balloon catheter [8]. The procedure was traditionally performed in the cardiac catheterization laboratory and assisted by uniplane fluoroscopy but was associated with an increased risk of complications. These risks decreased with the advent of biplane fluoroscopy, which instead led to an increase in ionizing radiation exposure. The introduction of two-dimensional transthoracic echocardiographic-assisted atrial septostomy (using the four-chamber subcostal or bicaval view) and the possibility of using the umbilical vein for the vascular approach have significantly simplified the procedure [9]. Nowadays, the intervention can be performed on the newborn’s bed in the neonatal intensive care unit (NICU), leading to a faster procedure, lower risk of traumatic injury to the atrioventricular valves, and reduced exposure time to ionizing radiation. The procedure can be performed under general anesthesia or mild sedation.

The Rashkind procedure can be performed using the femoral or umbilical vein (easy to approach in the newborn but difficult if the venous duct has a sinuous path) for vascular access. In cases where femoral or umbilical access is unavailable, such as femoral vein thrombosis, the transhepatic approach can be used, but with a reported rate of intraperitoneal bleeding of 4.5% even in experienced centers [10]. Internal jugular vein access is not routinely used because of the difficulty entering the left atrium via this route and the risk of injuring the sinoatrial node. However, several reports of successful BAS use this venous access [11].

A variety of catheters can be used to perform BAS. Currently, the most widely used are the 5F Miller catheter, which requires a 7F or 8F dilator for insertion, and the Z-5 double lumen catheter for atrial septostomy with variable sizes of 4F or 5F depending on the diameter of the balloon, which requires a dilator of 5F and 6F, respectively. Unlike the Miller catheter, the Z-5 catheter is advanced with a guide wire.

In the case of femoral access, the vein is punctured, and the septostomy catheter is placed in the vessel and then replaced with a dilator. The balloon catheter is advanced into the right atrium, penetrating the foramen ovale, and then the catheter is positioned in the left atrium. The correct position of the catheter is verified using biplane fluoroscopy or two-dimensional echocardiography. The balloon is inflated with 3–4 mL of dilute radiopaque agent (in case of fluoroscopic guidance) or saline solution (in case of echocardiographic guidance) and then rapidly withdrawn into the right atrium and deflated. The maneuver can be repeated two to three times. In the end, the results of the septostomy (size of the interatrial communication, absence of traumatic complications) are documented by echocardiography.

In the umbilical approach, the umbilical cord is sectioned tangentially to the skin, and the catheter is inserted directly into the umbilical vein using forceps. The progression of the catheter through the venous duct can be monitored by echocardiography or fluoroscopy. Sometimes it may be difficult for the catheter to pass into the inferior vena cava due to stenosis or closure of the ductus venosus. In this case, a guidewire approaches the umbilical vein and enters the right atrium. Then, a 7F or 8F sheath can insert the septostomy catheter. Still, the sheath must be withdrawn into the venous duct before the actual penetration of the interatrial septum begins. Once the catheter reaches the right atrium, the procedure continues, as described in the case of the femoral vein approach. 

### 1.3. Procedural Complications

Atrial septostomy is an invasive procedure that involves the inherent risks of any intravascular procedure. Prior planning of the procedure and universal antiseptic precautions are mandatory. Complications of the Rashkind procedure are rare, although important differences exist in reported rates. Complications of atrial septostomy [12] can be classified as described in the table below (Table 1).

## 2. Materials and Methods

### 2.1. Study Design

We conducted an observational, retrospective, single tertiary-care center study between January 2008 and April 2022, which included 92 neonates with TGA transferred to our institution for specialized treatment. This study excluded only one patient due to a complex diagnosis of the hypoplastic left heart.

Clinical data were retrospectively collected from hospital medical records. From each patient, we collected the following information: gender, gestational age, birth weight, Apgar score, age, time of cardiac diagnosis, comorbidities, echocardiographic measurements, respiratory support, inotropic and vasopressor support, prostaglandin E1, blood gas values, head ultrasound, balloon atrial septostomy technique, and cardiovascular surgical intervention. We divided patients into two groups according to the presence or absence of balloon atrial septostomy procedure.

The processing and statistical interpretation of the obtained data were carried out with the help of software Microsoft Excel 2007, SPSS Statistics 15.0.0 (SPSS Inc., Chicago, IL, USA), and MedCalc 14.8.1 (MedCalc software—2014).

### 2.2. Aim

Our three main objectives of this study were to assess the clinical benefit of the procedure, possible complications, and to determine whether the procedure should be performed in all patients with TGA with small atrial septal defect, regardless of oxygen saturation, in a center where corrective surgery cannot be performed on an emergency basis due to the lack of a permanent cardiac surgery team for arterial switch surgery.

### 2.3. Diagnosis and Procedure

The maximum estimated delay time for surgery was three weeks, considering the regular schedule of the cardiac surgery team in our center or the possibility of transferring the child to another cardiac center.

The overall severity of illness was assessed within the first 24 h of admission using a validated composite score to predict mortality and morbidity in the NICU—Score for Neonatal Acute Physiology with Perinatal Extension-II (SNAPPE II) [13].

Complex or simple TGA diagnosis was established by transthoracic echocardiography. All neonates were evaluated preoperatively by the neonatologist and pediatric cardiologist to assess the need for BAS and determine the optimal surgical approach.

The indication for balloon atrial septostomy was made based on clinical status (low systemic arterial oxygen saturation) correlated with tissue hypoxia (persistently elevated lactate level), echocardiographic assessment (patency and size of interatrial septal defect, ductus arteriosus, and interventricular septal defect), and anticipated delay to surgery. The procedure was performed according to the institutional protocol, at the patient’s bedside, under general anesthesia, and using the Rashkind pull-back technique under echocardiographic guidance (Figure 1 and Figure 2). Vascular access was obtained via a sterile technique via the femoral or umbilical vein. The procedure was repeated several times until satisfactory atrial communication was obtained. The success of BAS was clinically objectified by increasing systemic arterial saturation by at least 10% and echocardiographically objectified by increasing interatrial communication (Figure 3).

## 3. Results

During the past 14 years (January 2008–April 2022), there were 92 patients with transposition of great arteries admitted to our neonatal intensive care unit, of which 44 were managed exclusively in our department, and 48 patients were transferred to other hospitals for surgical correction of the cardiac malformation (Figure 4). Since 2012, 25 patients (27.1%) underwent balloon atrial septostomy: 19 neonates with TGA/IVS and 6 with TGA/VSD (2 with small VSD and 4 with moderate VSD). A detailed distribution by year and characteristics of patients with TGA admitted to our unit is presented in Table 2.

Most neonates had complex TGA (45.6% with associated VSD), while 38% had simple TGA. Moreover, nine patients (9.7%) had multiple congenital anomalies (anorectal malformation, liver hemangioma, pulmonary isomerism, agenesis of the corpus callosum, cerebellar hypoplasia, asplenia, situs ambiguus, choanal atresia, congenital ocular anomaly, and/or hypospadias).

The median age at admission was five days, with 21.7% of patients admitted on the first day of life, 39.1% transferred to our unit during the first week of life, 28.2% transferred during the first month of life, and 10.8% admitted after one month of age (Table 3).

Only 19.5% of patients were diagnosed prenatally, whereas the rest were diagnosed postnatally: 48.9% on the first day of life, 27.1% during the first week, and 4.3% after. An even smaller percentage of neonates who underwent the Rashkind procedure (15.3%) had a prenatal diagnosis of cardiac malformation.

Most patients (82.6%) were term neonates, but 16 were premature (13 late preterm, 2 moderate preterms, and 1 very preterm), with a median gestational age of 38 weeks and a median birth weight of 3170 g. Eleven patients had moderate hypoxia at birth, and one had severe hypoxia. Four patients who underwent BAS were premature (three late preterms and one moderate preterm) but none with perinatal hypoxia. A more detailed overview of demographics and hospitalization is illustrated in Table 3.

Most patients with TGA (69.5%) had optimal oxygenation at admission (SpO2 > 75% in both term and preterm babies). The overall group of patients with TGA had a greater median SpO2 at admission, as opposed to the group of newborns undergoing BAS (83%, respectively 78%) (Table 3).

Most TGA infants had a respiratory failure from the early days of admission: 61.9% of neonates required invasive respiratory support, 16.3% required noninvasive respiratory support, and 21.7% did not have respiratory failure. The percentage increased during hospitalization, with 76% requiring mechanical ventilation. On the other hand, all patients who underwent BAS procedures were oxygen-dependent at admission, with 88% requiring invasive ventilatory support (Table 3).

Cardiac insufficiency with a need for inotropic or vasopressor support was present in 41.3% of all TGA patients during the early days of admission and in an even larger percentage of patients in the BAS group (64%), with a median vasoactive inotropic score (VIS) of 7.5 (ranging from 0 to 105).

PGE1 infusion was necessary starting at admission in 82.6% of all TGA patients in various dosage regimens (median dose of 0.05 µg/kg/min). In contrast, all patients on whom we performed BAS required PGE1 infusion before the procedure (Table 4).

The median age at the moment of the Rashkind procedure was four days. Overall, out of 32 attempts, we considered the balloon atrial septostomy successful in 68.7% of cases (22). A second attempt was necessary in seven cases after the first attempt of the procedure was unsuccessful; in one case, even the second attempt was unsuccessful. In almost half of the cases (56.2%, 18), we used the femoral vein for vascular access with a superior rate of success (88.8%), in comparison with the umbilical vein (14 cases), with a 57.1% rate of success.

The rate of immediate complications after BAS was high (34.3%), but most were transient (metabolic acidosis and arterial hypotension—21.8%). Other complications included the following: heart rhythm disorders (supraventricular paroxysmal tachycardia)—9.3% and deep vein thrombosis—3.1%.

Twenty patients with TGA managed in our hospital underwent definitive and corrective surgical intervention (arterial switch operation) at a median age of 13 days. Meanwhile, three patients with complex TGA underwent palliative interventions.

The overall mortality of neonates with TGA in our clinic was 20.6%, with a preoperative mortality of 14.1% and a postoperative mortality of 26%. There was a significant difference in mortality between BAS patients (28%) and non-BAS patients (17.9%).

All TGA patients’ median length of stay was 15.5 days, ranging from 0 to 233 days, with a distinct difference between patients transferred from our unit to other clinics for surgical intervention (11 days) and patients operated in our clinic (34 days). The shortest hospitalization period for a patient who undertook an arterial switch operation in our unit was 13 days, while the longest was 233 days. Patients with optimal oxygenation at admission operated on before two weeks of life had a lower hospital stay duration (almost half of the stay in the BAS group and a third in the non-BAS group) (Table 5).

## 4. Discussion

Significant regional differences regarding managing newborns with transposition of great arteries exist. Without a large-scale prenatal diagnosis in Romania, newborns with TGA need to be stabilized at a regional hospital and transported to a tertiary-care center with expertise in managing congenital heart diseases where cardiovascular surgical care is available. Ideally, apart from initiating PGE1 infusion for maintaining ductal patency, atrial septostomy should be performed locally in selected cases of TGA, allowing for adequate shunting and stabilization prior to transport.

In our 14 years of experience, 92 neonates with TGA were hospitalized in our neonatal intensive care unit. There were an almost 2:1 ratio of male (63%) to female (36.9%) patients, in accordance with the double male predominance reported in other studies [14]. A similar distribution was present in the BAS group of neonates (60% boys and 40% girls).

The incidence of low-birth-weight newborns among patients with TGA was 8.7%, almost three times higher than the 3% reported incidence [5]. Low birth weight and prematurity are associated with technical and physiological perioperative challenges [15]. However, delaying the arterial switch operation to allow for weight gain is associated with higher morbidity and mortality [16]. Our cardiovascular surgical experience with this category of fragile newborns is limited, having only one low-birth-weight (2490 g) and four preterm infants (36 weeks of gestation) operated in our institution, with a rate of success of 80%. Nonetheless, four premature newborns underwent the Rashkind procedure (lowest gestational age of 33 weeks) and two low-birth-weight newborns (lowest weight of 1900 g), with all procedures being successful.

More than 80% of patients did not have a prenatal diagnosis. On average, they were transferred to our unit at a median age of 5 days after the suspicion of complex cardiac malformation was raised. This represents a major challenge since prenatal diagnosis of TGA facilitates the early introduction of PGE1 infusion, decreases the required time for BAS when indicated, and is shown to be efficient in reducing mortality and morbidity [17]. For these reasons, it is generally recommended for the delivery to occur at or near a tertiary-care pediatric cardiac surgery center, which enables optimal patient stabilization and avoids long-distance transport-related complications and costs.

Long-distance transport of neonates with TGA can be safe when performed by specialist transport teams, with a reported mortality of 0.04% [18]. During our 14 years of experience, we transported 48 neonates with TGA (at a median age of 21.5 days) to other cardiovascular surgery centers due to the inability to locally perform the surgical correction in due time. Almost all patients had severe general status at the time of the transport, with respiratory support (47.9% invasive, 45.8% noninvasive) and a median SpO2 of 85%; almost one in four had vasopressor and inotropic support (31.2%), and 81.2% had continuous infusion with PGE1.

Most TGA patients had a respiratory failure at admission (78.2%), necessitating respiratory support. One of the possible explanations for the severe general status of these babies during the first days of admission, with metabolic acidosis, tissue hypoxia, respiratory insufficiency, and need for ventilatory support, is the late postnatal diagnosis, leading to the closure of PDA in the absence of PGE1 initiation. Patients in the BAS group had an even more severe general status at admission, with a more decreased systemic arterial oxygenation than the non-BAS group (median SpO2 78% versus 84%). All BAS patients had a respiratory failure at admission and were oxygen-dependent. The percentage of patients with cardiac insufficiency on admission was double in the BAS group compared to the non-BAS group (64% vs. 32.8%). The median SNAPPE II score at admission was slightly higher (22.1) among neonates in the BAS group who died compared to 17.5 for those who survived. However, we found no correlation between the SNAPPE-II score at admission and the mortality of the patients who underwent the BAS procedure.

Intravenous PGE1 is routinely administered in patients with TGA for reopening or maintaining the patency of the ductus arteriosus. Most of our patients (82.6%) were administered various dosage regimens of PGE1 (median dose of 0.05 µg/kg/min), except for 16 patients with large PDA or associated CAVC, DORV, or tricuspid valve atresia with large VSD. Despite being necessary, the therapy has several dose-dependent side effects that need to be considered, including peripheral vasodilation leading to hypotension, apnea, fever, and tissue edema. Thus, the lowest effective dose is recommended to avoid complications. Starting doses for maintenance of ductal patency vary from 0.01 to 0.05 µg/kg/min. A higher dose of up to 0.1 µg/kg/min was necessary in 20 patients (21.7%) when the ductus needed to be reopened. Prolonged PGE1 administration was associated with cortical hyperostosis in one of our patients.

In several cases, using PGE1 may be insufficient in achieving optimal systemic arterial oxygenation in restrictive interatrial communication, entailing an atrial septostomy. After achieving adequate interatrial communication and shunting through the Rashkind procedure, PGE1 dosage can be gradually reduced but not interrupted due to the risk of rebound hypoxemia [19].

The balloon atrial septostomy in the NICU was performed at the patient’s bedside using echocardiographic guidance. This reduces further infant handling, avoids delay due to patient transfer to the cardiac catheterization laboratory, and is less expensive while remaining safe and efficient. We establish the need for clinical atrial communication enlargement via the Rashkind procedure and echocardiographic based on the dimension of ASD and ductus arteriosus.

The efficacy of balloon atrial septostomy is assessed by echocardiographic measurement of interatrial communication and clinical improvement in the patient’s condition. The success criteria for BAS can be variably defined in the literature as an increase of at least 10% in peripheral oxygen saturation [20] in the background of good atrial communication (increase in the ASD diameter > 1/3 of the total septal diameter) [21]. Based on these criteria, the Rashkind procedure was successful in 68.7% of our cases.

Femoral and umbilical veins are the access points of choice for the procedure. In 56.2% of the cases [20], we used the femoral vein for vascular access at a median age of 5 days. In the rest of the attempts (14 cases), umbilical access was used at a median age of 2 days. There was a clear difference in success depending on the vascular access of choice (88.8% success rate for femoral vein access versus 57.1% for umbilical vein access). The main reason for the failure of BAS when using umbilical access was the impermeability of the ductus venosus. At the same time, the failures of the femoral venous approach were attributed to the inability to penetrate the atrial septum.

Oxygenation improved in 76% of our patients after the Rashkind procedure, leading to an average increase of 9% in SpO2. Respiratory support could be decreased in 72% of cases after the BAS procedure, and FiO2 was reduced with an average of 13% in the following 6 h. The procedure also helped with the extubation of seven patients (28%) who were on assisted ventilation.

Hemodynamic improvement was also reported in most patients, which helped decrease the inotropic and vasopressor support. In the cases where BAS was unsuccessful, the vasoactive inotropic score increased during the following days. On the other hand, VIS decreased at 48 h post-BAS in 64% of cases where the procedure was successful, even though three patients required an increase of the inotropic-vasopressor support immediately after the procedure with a gradual reduction in the following hours.

Following the BAS procedure, we observed a significant decrease in lactate levels in 79.1% of cases, suggesting an improvement in systemic oxygenation and tissue perfusion. Nonetheless, in 63.1% of these patients, there was first an increase in lactate level followed by a steady reduction.

All patients received PGE1 when the Rashkind procedure was carried out (median dose of 0.1 µg/kg/min). In our experience, 80.7% of BAS procedures allowed for a reduction of PGE1 with a median of 0.04 µg//kg/min, while in the rest of the cases, the dosage remained unchanged.

The rate of immediate complications after BAS was high (34.3%), but most were transient (metabolic acidosis and arterial hypotension—21.8%). Other complications included the following: heart rhythm disorders (supraventricular paroxysmal tachycardia, ventricular fibrillation)—9.3%, and deep vein thrombosis—3.1% (without available information on the procoagulant status of the patient). We did not experience any mechanical or traumatic adverse events.

Twenty patients with TGA underwent definitive and corrective surgical interventions (arterial switch operation). There is no consensus over the exact optimal timing for surgery. Still, arterial switch operation should be performed within the first 3 weeks of life for TGA with intact IVS (Class I recommendation, level of evidence B in current European guidelines) [5]. In Europe, 25–30% of patients with simple TGA undergo routine arterial switch operations within the first week of life [7]. Performing the ASO during the first days of life is associated with better outcomes and reduced hospital costs, according to a retrospective study of 140 cases over 10 years, with age at the time of operation ranging from 1 to 12 days [22]. The risk of significant morbidity decreased by 46% for every day the surgery was delayed between the first and the third day of life. After three days of life, the risk increased by 47% for every day the surgical correction was delayed.

The median age at the time of operation was 13 days for the 23 patients operated on in our hospital. Almost one-third of the operated neonates (7) had the surgical intervention in the first week of life (100% survival rate), four patients in the second week of life (75% survival), and six during the third week of life (66.6% survival). Six patients were operated after 21 days of life (50% survival).

Some centers opt for a different strategy consisting of primary ASO on the first day of life, without BAS palliation, improving survival [23]. However, this treatment strategy is almost impossible in Romania, where most cases of TGA are diagnosed postnatally, and deliveries occur in non-tertiary-care centers, leading to a delay in initiating specific cardiovascular management. In this setting, early palliative BAS is followed by definitive surgery beyond the first week of life due to logistic reasons.

The overall mortality of neonates with TGA (operated and non-operated) in our clinic was 20.6%. The preoperative mortality of 14.1% is higher than that of newborns with TGA stated in several studies, ranging from 3.6% to 10.3% [24]. The unoperated deceased patients had a median age of 40 days, meaning surgery was delayed well beyond the recommended age. The preoperative mortality rate did not correlate with the oxygenation at admission but with the delay to surgery after 2 weeks of life. The reasons for the delay or not operating were variate: late presentation in critical status, severe sepsis, the impossibility of urgent transfer for surgical correction in another center (in settings of the absence of a local surgical team at that moment), or TGA in context of a very complex cardiac malformation or associated severe extra-cardiac pathology. The mortality of operated patients was 26%, with major differences depending on the operative time (9% mortality rate for patients on <14 days, as opposed to 41.6% in case of surgical correction beyond 14 days). Overall, 83.3% of deceased patients had been operated on after two weeks of life.

A recent study that included 17,392 neonates found no significant statistical difference in mortality between patients undergoing BAS and those without BAS [25]. However, BAS patients had a double increase in the length of stay and significantly higher rates of ECMO (5.1% vs. 3.1%) and stroke (1.1% vs. 0.6%). In our case, BAS patients had a 28% mortality rate, while for non-BAS patients, the mortality rate was 17.5% due to the more severe general status of neonates who underwent BAS (all BAS patients had a respiratory failure at admission as opposed to 78.2% in the non-BAS group; 64% of BAS patients had cardiac insufficiency versus 32.8%, respectively) and due to the delay to surgery (62.5% of BAS patients were operated after two weeks of life versus 46.6% of non-BAS patients).

BAS led to an improvement in preoperative survival in patients with optimal oxygenation at admission (100% survival rate before surgery in the BAS group, as opposed to 84% in non-BAS patients), as opposed to those with poor oxygenation (preoperative mortality of 36.3% versus 0% in the non-BAS group). We hypothesize this means that BAS proves even more efficient when performed beyond the recommended indication to those with good oxygen saturation and non-large ASD when there is a delay to surgery beyond the first three days of life.

Some neonates may remain significantly cyanotic and acidotic even after the atrial septostomy, in case of persistent pulmonary hypertension of the newborn (PPHN), present in up to 12% of neonates with TGA [26]. However, we found a smaller incidence of pulmonary hypertension in our patients (8.6%), with no difference between patients undergoing BAS and not. Patients with TGA and PPHN have poor outcomes: longer ventilatory support and NICU stay, more frequent need for ECMO, and preoperative mortality can reach up to 29% [27].

We had a single patient who required extracorporeal membrane oxygenation (ECMO) in the immediate postoperative time due to inadequate hemodynamics despite high inotropic and vasopressor support, progressive lactic acidosis, and progressive increase in tissue perfusion markers. The patient had undergone a balloon atrial septostomy procedure with success at nine days of life; unfortunately, because of severe sepsis, the surgery had to be postponed until the 36th day of life, resulting in the patient’s death on the second postoperative day. The need for extracorporeal life support is reported in around 20% of infants undergoing arterial switch operation beyond the sixth week of life [28].

Patients with TGA have a more favorable neurodevelopmental outcome than those with other cyanotic cardiac malformations [29]. However, they may still develop motor dysfunction, learning disabilities, and behavioral disorders due to prolonged low oxygenation of the blood delivered to the brain. Nine of our patients with TGA (9.7%) developed neurological lesions during hospitalization (intraventricular hemorrhage, hydrocephalus, seizures). Brain injuries were previously thought to be caused by cardiopulmonary bypass or the displacement of pre-existing thrombi during balloon atrial septostomy. However, recent studies point towards a multifactorial cause. Preoperative brain injuries are correlated with hypoxemia and delay to surgery, rather than BAS, suggesting a relationship between brain vulnerability, postnatal lower arterial oxygen saturations, greater incidence of metabolic acidosis, and neurological complications [30]. The incidence of neurologic complications in the BAS group was higher (12%) compared to the non-BAS group (8.9%). A possible explanation is that neonates with TGA and restrictive atrial communication who require BAS have more severe hypoxemia and hemodynamic instability, which increases the likelihood of an ischemic or hemorrhagic event.

## 5. Limitations

This was a retrospective, single-center study carried out over a relatively long period, with limited initial experience managing patients with congenital heart defects requiring complex surgery. In addition, cardiovascular procedures for patients with TGA were performed monthly due to the non-permanent cardiac surgery team for this type of intervention. This could have affected outcomes, with significant differences during the training years.

## 6. Conclusions

In Romania, a small proportion of neonates with cardiac malformations are diagnosed prenatally, delaying diagnosis and initiating adequate cardiovascular management.

This study presents evidence for the inclusion of BAS in the standard management protocol for neonates with TGA (with extended indication to those with good oxygen saturation and non-large ASD) in neonatal intensive care units of medical centers where surgical correction is expected to be delayed beyond the first three days of life but not beyond two weeks due to non-permanent access to cardiac surgery for this type of intervention or other logistical reasons. In these circumstances, more studies are needed on the benefits of an extended BAS and the optimal preoperative strategy for neonates with TGA.

## Figures and Tables

**Figure 1 life-13-00997-f001:**
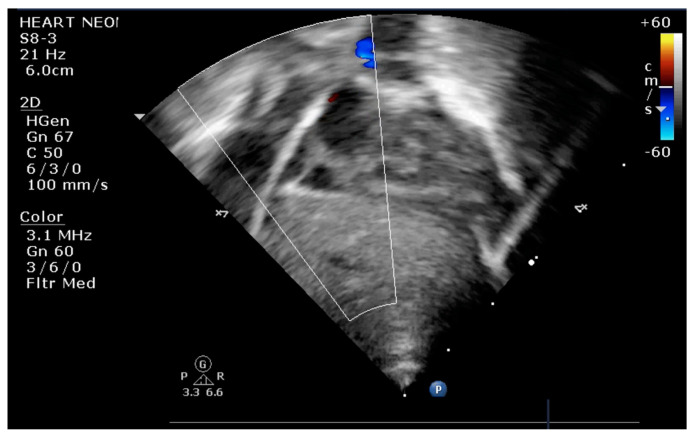
Transthoracic echocardiography (TTE) subcostal view: atrial septostomy catheter in the LA.

**Figure 2 life-13-00997-f002:**
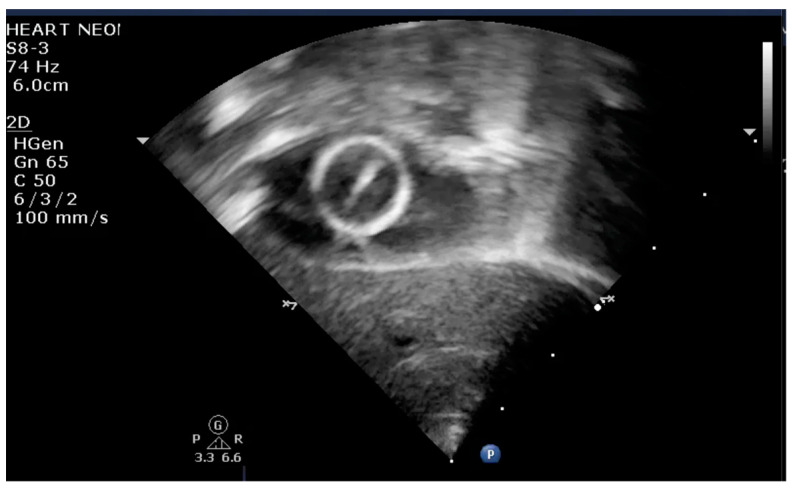
Transthoracic echocardiography subcostal view: inflated balloon in the LA.

**Figure 3 life-13-00997-f003:**
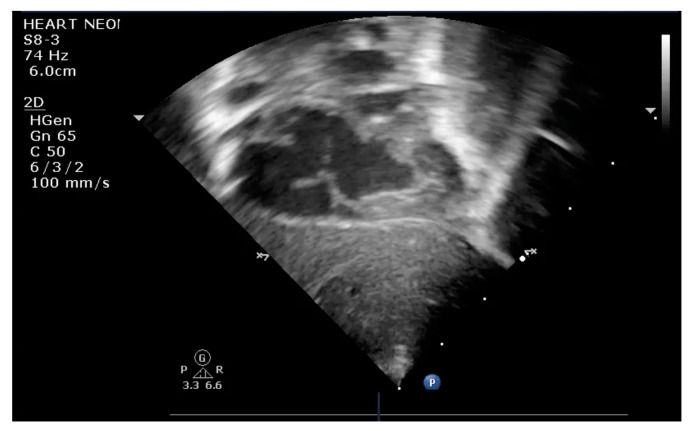
Transthoracic echocardiography subcostal view: large communication at the level of IAS after BAS.

**Figure 4 life-13-00997-f004:**
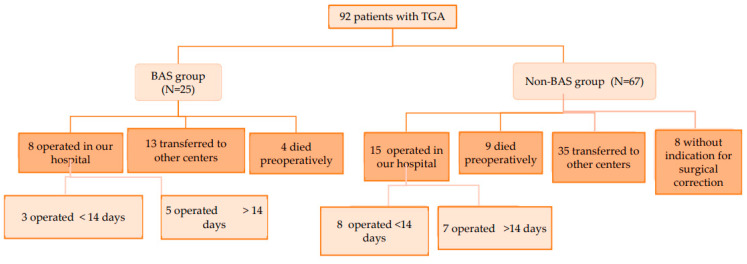
Study design.

**Table 1 life-13-00997-t001:** Complications of atrial septostomy.

Mechanical	Traumatic	Embolic	Electrophysiological
Rupture of the balloon	Cardiac damage by rupture of LAA	Cerebral ischemia	Transitory heart rhythm disorders
Balloon deflation failure	Mitral valve damage		
Inflation of the balloon in an inappropriate position	Perforation of pulmonary veins of IVC		

**Table 2 life-13-00997-t002:** Distribution by year and characteristics of patients with TGA admitted in our unit. CHD—congenital heart disease.

Year	Patients with TGA (N = 92)	Patients Undergoing BAS (N = 25)	Surgical Correction in Our Hospital (N = 23)	Preoperatory Survival	Postoperatory Survival	Referral to Other Centers for Surgical Intervention (N = 48)	Overall Survival
2008	4	0	0	3 (75%)		3 (75%)	3 (75%)
2009	0						
2010	3	0	0	2 (66.6%)		0	2 (66.6%)
2011	2	0	0	1 (50%)		1 (50%)	1 (50%)
2012	5	2 (20%)	0	3 (60%)		3 (60%)	3 (60%)
2013	3	1 (33.3%)	0	3 (100%)		2 (66.6%)	3 (100%)
2014	4	0	0	4 (100%)		4 (100%)	4 (100%)
2015	7	0	2 (28.5%)	3 (42.8%)	2 (100%)	1 (14.2%)	3 (42.8%)
2016	15	4 (26.6%)	4 (26.6%)	14 (93.3%)	4 (100%)	1 (6.6%)	14 (93.3%)
2017	10	4 (40%)	1 (10%)	9 (90%)	1 (100%)	7 (70%)	9 (90%)
2018	11	4 (36.3%)	3 (27.2%)	10 (90.9%)	2 (66.6%)	5 (45.4%)	9 (81.8%)
2019	6	1 (16.6%)	0	6 (100%)		5 (83.3%)	6 (100%)
2020	13	4 (30.7%)	8 (61.5%)	13 (100%)	5 (62.5%)	5 (38.4%)	10 (76.9%)
2021	5	2 (40%)	3 (60%)	4 (80%)	1 (33.3%)	1 (20%)	2 (40%)
2022	4	3 (75%)	2 (50%)	4 (100%)	100%	2 (50%)	4 (100%)

**Table 3 life-13-00997-t003:** Demographics and hospitalization characteristics.

	Overall TGA Patients (N = 92)	non-BAS Group (N = 67)	BAS Group (N = 25)
Gender			
Female	34 (36.9%)	24 (35.8%)	10 (40%)
Male	58 (63%)	43 (64.1%)	15 (60%)
Age in days at presentation—median	5	8	3
Gestational age—median	38	38	38
Term	76 (82.6%)	55 (82%)	21 (84%)
Preterm	16 (17.3%)	12 (17.9%)	4 (16%)
Birth weight—median (grams)	3170	3200	3150
Apgar score—median	8	8	8
Oxygenation at admission			
Median SpO2	83%	84%	78%
SpO2 > 75%	64 (69.5%)	50 (74.6%)	14 (56%)
Respiratory support at admission			
None	20 (21.7%)	20 (29.8%)	0
Noninvasive	15 (16.3%)	12 (17.9%)	3 (12%)
Invasive	57 (61.9%)	35 (52.2%)	22 (88%)
Inotropic and/or vasopressor support	38 (41.3%)	22 (32.8%)	16 (64%)
Adrenaline	13 (14.1%)	8 (11.9%)	5 (20%)
Noradrenaline	5 (5.4%)	3 (4.4%)	2 (8%)
Dopamine	23 (25%)	14 (20.8%)	9 (36%)
Dobutamine	16 (17.3%)	10 (1.4%)	7 (28%)
Milrinone	7 (7.6%)	6 (8.9%)	1 (4%)

**Table 4 life-13-00997-t004:** Characteristics of operated patients with optimal oxygenation at admission.

TGA Patients (N = 92)		Preoperative Mortality
Patients with SpO2 > 75% (N = 64)	BAS patients (N = 14)	0%
	non-BAS patients (N = 50)	8 (16%)
Patients with SpO2 < 75% (N = 28)	BAS patients (N = 11)	4 (36.3%)
	non-BAS patients (N = 17)	0%
Patients with SpO2 > 75%		Hospital Stay (Days)
BAS patients (N = 14)	Operated < 14 days (N = 2)	42
	Operated > 14 days (N = 5)	76.2
non-BAS patients (N = 50)	Operated < 14 days (N = 4)	23.7
	Operated > 14 days (N = 5)	90

**Table 5 life-13-00997-t005:** Associated comorbidities.

	Overall TGA Patients (N = 92)	non-BAS Group (N = 67)	BAS Group (N = 25)
Cardiac			
PDA	82 (89.1%)	60 (89.5%)	22 (88%)
small (<1.5 mm)	2 (2.1%)	2 (2.9%)	0
moderate (1.5–3 mm)	20 (21.7%)	15 (22.3%)	5 (20%)
large (>3 mm)	60 (65.2%)	43 (64.1%)	17 (68%)
ASD	90 (97.8%)	65 (97%)	25 (100%)
small (<6 mm)	69 (75%)	47 (70.1%)	22 (88%)
moderate (6–12 mm)	17 (18.4%)	14 (20.8%)	3 (12%)
large (≥12 mm)	4 (4.3%)	4 (5.9%)	0
VSD	42 (45.6%)	36 (53.7%)	6 (24%)
small (≤3 mm)	11 (11.9%)	9 (13.4%)	2 (8%)
moderate (3–6 mm)	19 (20.6%)	15 (22.3%)	4 (16%)
large (>6 mm)	12 (13%)	12 (17.9%)	0
Aortic coarctation and/or aortic arch hypoplasia	15 (16.3%)	13 (19.4%)	2 (8%)
DORV	11 11.9%	11 (16.4%)	0
CAVC	8 (8.6%)	8 (11.9%)	0
TAPVR	6 (6.5%)	6 (8.9%)	0
Pulmonary atresia	8 (8.6%)	8 (11.9%)	0
Tricuspid atresia	7 (7.6%)	7 (10.4%)	0
Pulmonary hypertension	8 (8.6%)	6 (8.9%)	2 (8%)
Respiratory			
Atelectasis	5 (5.4%)	3 (4.4%)	2 (8%)
Pleural Effusion	5 (5.4%)	2 (2.9%)	3 (12%)
Pneumothorax	3 (3.2%)	2 (2.9%)	1 (4%)
Pulmonary hemorrhage	2 (2.1%)	1 (1.4%)	1 (4%)
SIRS	19 (20.6%)	12 (17.9%)	7 (28%)
Sepsis	7 (7.6%)	5 (7.4%)	2 (8%)
Acute kidney injury	8 (8.6%)	5 (7.4%)	3 (12%)
Hemodiafiltration	1 (1%)	0	1 (4%)
Hepatic failure	5 (5.4%)	3 (4.4%)	2 (8%)
Gastrointestinal			
Upper digestive hemorrhage	2 (2.1%)	1 (1.4%)	1 (4%)
Neurological			
Intraventricular hemorrhage	5 (5.4%)	3 (4.4%)	2 (8%)
Post-hemorrhagic hydrocephalus	3 (3.2%)	1 (1.4%)	2 (8%)
Seizures	5 (5.4%)	4 (5.9%)	1 (4%)

PDA: Patent Ductus Arteriosus; ASD: Atrial Septal Defect; VSD: Ventricular Septal Defect; DORV: Double Outlet Right Ventricle; CAVC: Complete Atrioventricular Canal Defect; TAPVR: Total Anomalous Pulmonary Venous Return.

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
