# Peer review of "Impact of Bedside Balloon Atrial Septostomy in Neonates with Transposition of the Great Arteries in a Neonatal Intensive Care Unit in Romania"

_life, 2023, doi:10.3390/life13040997_

Round 1

Reviewer 1 Report

Dear Editor and Authors,

It was a pleasure to read and review this manuscript titled “Impact of Bedside Balloon Atrial Septostomy in Neonates with Transposition of the Great Arteries in a Neonatal Intensive Care Unit in Romania” by Dr. Cirstoveanu and colleagues from “Carol Davila” University of Medicine and Pharmacy in Bucharest, Romania.

In this retrospective single institution observational case control analysis the authors report their experience with balloon atrial septostomy (BAS) performed during the first week of life so that infants with transposition of the great arteries can be stabilized pre-operatively. Data of 92 neonates in total over a 14-year period are analyzed, compared and presented. Twenty five infants underwent BAS whereas 67 did not and those comprised the study and control groups respectively. The authors report that the balloon atrial septostomy was successful in 68,7% of cases! The rate of complications after BAS was high (34,3%), but transient.

Overall this is quite a nice paper. It is well written in a clear and well structured language with only minor linguistic mistakes so all it needs is a quick review by a native English speaker or a professional service if possible. It is concise and comprehensive and it is well illustrated with clear tables and graphs and informative figures.

I have only a couple of minor suggestions to improve the work:

1. In table 4 please change feminine and masculine to male and female.

2. I would not call table 2 a table but rather a graph or a figure. As such please rename the tables and figures.

3. Why were patients transferred to other units to be operated on? What was the reason since it appears your hospital / unit has the expertise to surgically manage these babies? I ask as a cardiothoracic surgeon because you are a tertiary university hospital!!

4. Can the authors explain why they have such a significant percentage of renal failure in the group overall?

5. In the discussion the authors mention that their “cardiovascular surgical experience with this category of fragile newborns is limited, having only one low birth weight (2490 g) and four preterm infants (36 weeks of gestation) operated in their institution” however in table 2 they report that 23 babies were operated in their center. Then on line 418 in the discussion they mention the number as been 20!! Which is correct? Please clarify.

In conclusion and as previously mentioned, this is a nice report and only needs some minor corrections before it can be accepted for publication. Thank you for giving me the opportunity to review this work. Wishing well to all.

Author Response

Dear reviewer,

We are grateful for your kind words regarding our work. We have addressed all of your comments and suggestions.

R1:

Dear Editor and Authors,

It was a pleasure to read and review this manuscript titled “Impact of Bedside Balloon Atrial Septostomy in Neonates with Transposition of the Great Arteries in a Neonatal Intensive Care Unit in Romania” by Dr. Cirstoveanu and colleagues from “Carol Davila” University of Medicine and Pharmacy in Bucharest, Romania.

In this retrospective single institution observational case control analysis the authors report their experience with balloon atrial septostomy (BAS) performed during the first week of life so that infants with transposition of the great arteries can be stabilized pre-operatively. Data of 92 neonates in total over a 14-year period are analyzed, compared and presented. Twenty five infants underwent BAS whereas 67 did not and those comprised the study and control groups respectively. The authors report that the balloon atrial septostomy was successful in 68,7% of cases! The rate of complications after BAS was high (34,3%), but transient.

Overall this is quite a nice paper. It is well written in a clear and well structured language with only minor linguistic mistakes so all it needs is a quick review by a native English speaker or a professional service if possible. It is concise and comprehensive and it is well illustrated with clear tables and graphs and informative figures.

I have only a couple of minor suggestions to improve the work:

  1. In table 4 please change feminine and masculine to male and female.

Thank you for this suggestion. We modified it accordingly.

  1. I would not call table 2 a table but rather a graph or a figure. As such please rename the tables and figures.

We renamed all the figures and tables properly.

  1. Why were patients transferred to other units to be operated on? What was the reason since it appears your hospital / unit has the expertise to surgically manage these babies? I ask as a cardiothoracic surgeon because you are a tertiary university hospital!!

Cardiovascular surgical interventions started taking place in our hospital in 2015. At the beginning of the learning curve, higher RACHS-1 risk procedures were referred to more experienced centers. Since 2020 more than half of the TGA patients have been operated in our clinic and only more complex cases were transferred to other centers for surgical intervention.

  1. Can the authors explain why they have such a significant percentage of renal failure in the group overall?

Eight patients (8,6%) developed acute kidney injury. Five of them had complex cardiac malformations (TGA associated with DORV, CAVC and/or aortic arch hypoplasia). Five of the acute kidney injuries occurred post-cardiac surgery (21,7% of the operated patients in our unit), one of them requiring continuous renal replacement therapy. Our results are in accordance with a recent retrospective review from a tertiary children’s hospital in the Netherlands, where 18% of patients developed AKI after arterial switch operation, and 5% developed renal failure[1].

  1. In the discussion the authors mention that their “cardiovascular surgical experience with this category of fragile newborns is limited, having only one low birth weight (2490 g) and four preterm infants (36 weeks of gestation) operated in their institution” however in table 2 they report that 23 babies were operated in their center. Then on line 418 in the discussion they mention the number as being 20!! Which is correct? Please clarify.

Twenty-three patients were operated on in our unit (20 underwent arterial switch operation, while 3 underwent palliative interventions - Blalock-Taussig shunt or pulmonary artery banding and atrial septostomy).

Our experience regarding cardiovascular surgical interventions in premature and low birth weight infants is limited.

In conclusion and as previously mentioned, this is a nice report and only needs some minor corrections before it can be accepted for publication. Thank you for giving me the opportunity to review this work. Wishing well to all.

Thank you for all the useful suggestions.

Kind regards,

The authors

Reviewer 2 Report

Dear authors,

This is an interesting study of performing the balloon atrial septostomy in the NICU, at the patient's bedside, in neonates with TGA. I have some comments and recommendations.

Abstract - Please specify in the first two paragraphs which specific procedure your study investigates. I know this can be deduced from the title, but I think it is necessary for the Abstract to stand on its own.

-The sentence describing the purpose of the study is too long and loses its meaning; please divide it into two.

-Line 38 - Please define BAS (first appearance in the text)

Introduction

-Line 70, 75 and 106-  Please define D-TGA, PFO and CHD

- Line 116-128 - This paragraph needs some references

Methods - this chapter needs a major revision.

Figure 1,2,3 - Please define TTE

- Please specify what kind of personal data were collected and analyzed in your cohort

-Please specify what kind of statistical methods you applied in the conducted study

Results

- Line 293-294 - What statistical test did you applied for this "significant" difference in mortality?

- Table 6 - Please define under the table all the abbreviation used

Discussion

-Line 315-316 - A reference is needed

-More comparisons with similar recent studies are needed, and also a paragraph defining the limitations of the study you performed.

Conclusion - this chapter is too wordy. 

One or two paragraphs summarizing the results of the study would be enough.

Author Response

Dear reviewer,

We are grateful for your kind words regarding our work. We have addressed all your comments and suggestions.

R2:

Dear authors,

This is an interesting study of performing the balloon atrial septostomy in the NICU, at the patient's bedside, in neonates with TGA. I have some comments and recommendations.

Abstract - Please specify in the first two paragraphs which specific procedure your study investigates. I know this can be deduced from the title, but I think it is necessary for the Abstract to stand on its own.

We have improved our abstract presentation.

-The sentence describing the purpose of the study is too long and loses its meaning; please divide it into two.

We divided the sentence following your recommendations.

-Line 38 - Please define BAS (first appearance in the text)

We defined all the acronyms.

Introduction

-Line 70, 75 and 106-  Please define D-TGA, PFO and CHD

We added all the definitions missing.

- Line 116-128 - This paragraph needs some references

We added two references.

Methods - this chapter needs a major revision.

Figure 1,2,3 - Please define TTE

We defined the TTE.

- Please specify what kind of personal data were collected and analyzed in your cohort

We added the missing information.

-Please specify what kind of statistical methods you applied in the conducted study

We added the missing data.

Results

- Line 293-294 - What statistical test did you applied for this "significant" difference in mortality?

We rephrased the sentence because the difference in mortality was significantly higher, not statistically significant.

- Table 6 - Please define under the table all the abbreviation used

We added the definitions of all abbreviations under the table.

Discussion

-Line 315-316 - A reference is needed

“A similar distribution was present in the Rashkind group of neonates (60% boys and 40% girls)“ - the Rashkind group is the BAS group from our current study. We renamed it “BAS group“ for a more clear understanding.

-More comparisons with similar recent studies are needed, and also a paragraph defining the limitations of the study you performed.    

We added a paragraph describing the study's limitations.    

Conclusion - this chapter is too wordy.

One or two paragraphs summarizing the results of the study would be enough.

We rephrased this paragraph.

Kind regards,

The authors

Round 2

Reviewer 2 Report

Dear authors,

Thank you for your answers. The manuscript is improved.

I only have a comment:

1. -"Line 293-294 - What statistical test did you applied for this "significant" difference in mortality?  Your response: We rephrased the sentence because the difference in mortality was significantly higher, not statistically significant." 

In order to be able to state that the difference in the mortality rate between the two groups was "significantly" higher, this fact must be demonstrated by applying a statistical test comparing proportions (for example, the z test). Otherwise, it can only be stated that for the respective groups of patients the mortality in one group is higher than in the other group.

Kind regards,

Author Response

Dear reviewer,

Thank you for pointing out the typo. We have rephrased it accordingly. "In our case, patients with BAS had a mortality rate of 28%, while for non-BAS patients, the mortality rate was 17.5%, due to the more severe general condition of the newborns who underwent BAS...."

Kind regards, 

The authors